# Multi-Scale Characteristics of Investor Sentiment Transmission Based on Wavelet, Transfer Entropy and Network Analysis

**DOI:** 10.3390/e24121786

**Published:** 2022-12-06

**Authors:** Muye Han, Jinsheng Zhou

**Affiliations:** School of Economics and Management, China University of Geosciences, Beijing 100083, China

**Keywords:** investor sentiment transmission, wavelet, transfer entropy, network analysis

## Abstract

Investor sentiment transmission is significantly influential over financial markets. Prior studies do not reach a consensus about the multi-scale transmission patterns of investor sentiment. Our study proposed a composite set of methods based on wavelet, transfer entropy, and network analysis to explore the transmission patterns of investor sentiment among firms. By taking 137 new energy vehicle-related listed firms as an example, the results show three key findings: (1) the transmission of investor sentiment presents more active in the short term and takes place in a local range; (2) the transmission of investor sentiment presents patterns of continuity and growth from short term to long term; and (3) the transmission patterns of investor sentiment will have specific evolutions from short term to long term. Suggestions are offered to investors, managers and policymakers to better monitor the financial market using investor sentiment transmission.

## 1. Introduction

Investor sentiment has aroused rising attention from financial investors and literature for its broad and multiplex impact on financial markets. Due to social media and market integration, investors’ emotions are easily propagated across populations and countries, which then forms a transmission network [1,2]. Literature has found that investor sentiment is transmissible across investors, firms, and markets [3,4]. As a result, local volatility of investor sentiment propagates quickly along the network and causes systemic impacts. Furthermore, since the limit to short arbitrage, the markets are dominated by optimistic investors but do not reflect sufficient pessimistic emotions, which causes the markets to become overheated. It is the trigger for most historical financial crises. In addition, once the bubbles burst, the panic quickly spreads among investors and lead to a market crash, such as the bank run and underselling [5]. Thus, it is vital to explore the transmission patterns of investor sentiment and understand its characteristics in order to monitor the market shock.

Among research on investor sentiment transmission, the spillover effect makes investor sentiments of different firms interactive, and the effect of multiple temporal scales makes the interactions complex [6,7,8]. To explore those effects, prior literature tends to use financial data to reflect the multi-scale effect [9], and use correlation measurements or agent-model-based simulations to abstract the sentiment transmission [10], few have constructed a composite set of methods and tools according to the complex interactiveness and multi-scale of investor sentiment transmission.

Specifically, first, the emotions and attitudes of investors toward the market prospect diffuse through online comments and analyst reports, which makes individual emotions become population sentiment. In addition, globalization, production specialization and complex competition–cooperation relationships among market participants also make firms around the world interdependent. The sentiment of a certain stock will influence its price, which is then observed by more investors and affects their attitudes toward the stock. Furthermore, the sentiment toward a certain asset will affect the investors’ attitudes towards other assets and derivatives due to interdependencies among firms, industries, and markets [11]. Those factors jointly result in the interactiveness of investor sentiment transmission. To well represent and investigate such interactiveness, complex network analysis is useful, which admits the complex interactions among investors, firms, and markets and constructs the topological structure of sentiment transmission.

Second, investors are the unity of the “animal spirit” and “rational-economic man” [12,13,14]. The characteristics of irrational emotions are often transmitted by violent fluctuations in a short time. Rational emotions represent investors’ long-term understanding of the investment objects. This rational understanding has difficulties in undergoing drastic changes, such as irrational emotions, in a short time. Instead, it evolves and propagates gradually over a long time scale. In addition, the short-term irrational emotion induced by emergencies and bad news will be gradually modified by the long-term rational emotions induced by aggregate economic fundamentals [15]. The coexistence of irrational and rational emotions leads to different transmission modes of investor sentiment on different time scales. This gives rise to the second attribute of investor sentiment transmission, i.e., multi-scale. From the time scale, investors can be generally divided into three categories: high-frequency traders who pay attention to very short-term strategies, technical investors who make profits through medium-term trading, and long-term investors who focus on long-term value investment [16]. Investors with different time scales will have different expectations for identical market information and reflect different emotions, which further leads to the different diffusion and transmission of investor sentiment on different time scales [17]. Thus, exploring the multi-scale differences in investor sentiment transmission can help to build an understanding of the intensity and patterns of rational and irrational emotion transmission in the short and long term.

Therefore, considering the above attributes of investor sentiment transmission and the lack of literature addressing these attributes, our study aims at deciphering the differences in investor sentiment transmission characteristics in the short term and long term through a composite set of methods. To figure it out, we select the Baidu search index for listed companies in China to represent investor sentiment. The Baidu search index is widely accepted to measure investor sentiment in the Chinese market [18]. First, considering the multi-scale features of investor sentiment transmission, we apply the wavelet method to decompose and extract the multi-scale information of investor sentiment for every listed company. Second, we construct the investor sentiment transmission networks (ISTNs) in the short and long term based on the transfer entropy among the investor sentiment for different listed companies. The ISTN takes into account the interactiveness of investor sentiment transmission not only from close neighbours but also from the whole network, while the transfer entropy method differentiates well between the ‘driving’ nodes and the ‘responding’ nodes, and is then able to measure the influences that one variable can have on another [19]. Third, we analyse and discuss the investor sentiment transmission characteristics in the short and long term based on network analysis to reveal how the investor sentiment transmit among listed companies on different scales and what the differences are in transmission characteristics between the short term and long term. The results exhibit the following main findings: (1) the transmission of investor sentiment presents more active in the short term and takes place in a local range; (2) the transmission of investor sentiment presents patterns of continuity and growth from short term to long term; (3) the transmission patterns of investor sentiment will have specific evolutions from short term to long term.

Our study contributes to the investor sentiment transmission literature in the following aspects. We apply a composite set of methods combining the wavelet, transfer entropy and complex network analysis. These methods are selected and integrated based on the multi-scale, complex interactiveness and directionality of investor sentiment transmission. This composite method can help to build a realistic transmission network of investor sentiment and uncover the local and network effect at multi-frequencies. In addition, this study is one of the few exploring the transmission patterns and characteristics of firm-specific investor sentiment. As far as we know, literature mainly focuses on the relationships between investor sentiment and asset returns and ignores that investor sentiment itself is intrinsically a complex information interactive system. Our findings can provide strategic suggestions for heterogeneous investors and market regulators on paying attention to specific investor sentiment according to short-term or long-term objectives.

The rest of this paper is organized as follows: Section 2 is the review of related works. Section 3 introduces and describes the data and methods applied. In Section 4, the experiment results and corresponding discussion are presented. In Section 5, the conclusion, strategic suggestions, and research perspectives are presented.

## 2. Literature Review

### 2.1. Investor Sentiment Transmission

Investor sentiment transmission is the process that individual investors, actively or passively, capture the sentiments of others and, thus, realize the interaction and aggregation of opinions and attitudes toward assets among different individuals [10]. While the sentiment of one stock will propagate among its potential and current holders, it can also be transmitted to other stocks due to the interdependency of business, which induces investor sentiment transmission across firms. After the financial crisis in 2008, investor sentiment has been an important factor for investors and academia to consider. Several studies have been performed in order to explore the characteristics of investor sentiment across investors, firms, and markets [9,20,21]. However, since investor sentiment is transmissible, which is interactive and multi-time-scale, there lacks a composite set of methods to explore the topic.

As for the studies on investor sentiment transmission, prior research use correlation and causality analysis to investigate the topic in the cross-market context. For example, Audrino and Tetereva [22] use Granger-causality to measure the spillover effect of news sentiment of stocks in one industry on the excessive return of stocks in another industry. They discuss the spillover relationships from the network perspective and find the financial and energy sectors have the most influential sentiment in terms of sentiment contagion. Niţoi and Pochea [23] adopt a copula-based model to explore the spillover effect of investor sentiment contagion among European equity markets. They argue that sentiment contagion increases the correlation of dissimilar markets. Tsai [24] adopts a cross-investor angle and constructs a dynamic sentiment spillover index to explore the diffusion of investor sentiment among dissimilar investors. They followed the method by Diebold and Yilmaz [25] to measure variable correlations and proxy the diffusions of sentiment. The results demonstrate the asymmetry of investor sentiment that the pessimistic view propagates faster than the optimistic sentiment.

As for the measurement of investor sentiment, previous studies applied a synthesizing method. For example, Baker et al. [3] use the principal component analysis to synthesize a composite index of investor sentiment and separate the total sentiment into global sentiment and six local sentiments. By regression analysis, they find that both the global and local sentiment has an impact on stock price, and the cross-market sentiment contagion partially results in the global sentiment. While recently, a social media source of sentiment has caught the attention of present researchers. For example, Gao et al. [20] use the Google search index to proxy investor sentiment of 38 countries from 2004 to 2014 and validate the effectiveness of the proxy using sports outcomes that sentiment is a contrarian predictor of market returns. Mendoza-Urdiales et al. [26] collect the top tweets of the 24 largest publicly traded companies on the social media site Twitter for 10 years and use semantic analysis to measure investor sentiment. Through the transfer entropy and EGARCH methods, they build the directed information transmission relationships between the firm-specific sentiment and stock prices. The results demonstrate that negative sentiment has a larger impact on stock performance. Compared to the synthesizing method that indirectly measures investor sentiment through financial and operational data, using the social media source directly represents investor sentiment.

However, the above review of related works has reflected the fact that few have investigated the interactiveness of investor sentiment transmission. The sentiment of different stocks or firms will be mutually influenced so it is necessary to consider the direction of transmission rather than a correlation without directions. In addition, prior studies mostly consider the transmission from ‘neighbours’. However, a firm’s investor sentiment will not only be influenced by its most-correlated partners and competitors, but also by parties from the entire business network. Thus, complex network analysis is an appropriate way to explore investor sentiment transmission.

### 2.2. Investor Sentiment Transmission Network and Transfer Entropy

As public opinion or information transmission exhibits a network form, a complex network method has been adopted in studying the process of sentiment transmission. As real-world complex systems can be described by complex networks, public opinion propagation in the form of networks is an effective tool to model investor sentiment transmission. For example, Zhao et al. [27] consider a 2-state emotional contagion model to describe the interactions of sentiment propagation and simulate the propagation process based on a Barabási–Albert scale-free network. The results show that the process of sentiment propagation is affected by close contact with neighbours and external general information. Direct contact represents the propagation process through close relationships and the external general information reflects the overall tendency of the whole external environment. In our study, we construct the investor sentiment transmission network to investigate the transmission characteristics of investor sentiment among firms.

The investor sentiment transmission network (ISTN) can be described as a network of interconnected entities, such as firms or stocks, in which entities act as nodes, and the interconnections between nodes act as edges [28]. The edges are the investor sentiment transmission relationships between any two firms. Different from physical networks driven by the real transactional relationships between firms, ISTN is driven by information transmission, such as spillover effect, correlation, and causality, which is data-driven and called a correlation network [29]. Thus, the measurement of the transmission relationships is the basis to build the ISTN. In addition, although ISTN is called a correlation network, it is insufficient to just measure the inter-correlations among firms.

In our study, the network connectivity is based on the estimation of the sentiment influence that one firm *i* transmits to another firm *j*. If significant sentiment transmission exists, there will be a directed edge from *i* to *j*. Based on this consideration, measurements of correlation are not appropriate since they are only applicable to determine the degree of relationship between the variables but do not provide the direction of the relationship [29]. Specifically, a correlation-based network can only be an undirected weighted network, which makes directed network topological analysis infeasible. Thus, measurements with directivity are preferred to build a transmission network, such as Granger-causality [29,30,31].

However, the relationships and interactions in financial markets are deemed to be non-linear, which makes Granger-causality, a linear measurement, not applicable. To deal with directivity and non-linearity, our study uses transfer entropy to measure the directed sentiment transmission relationships. Transfer entropy is the measurement of information transfer which distinguishes the source and target of information flow first introduced by Schreiber [32] in 2020. It can measure the asymmetric information flow between variables in a model-free manner, which makes it applicable to analyse non-linear complex systems, such as ISTN [19,33,34]. By calculating the transfer entropy from one firm *i* to another firm *j*, we can not only capture the direction of sentiment transmission but also the degree and strength of this transmission relationship, which builds a directed-weighted complex network.

### 2.3. Multi-Scale Investor Sentiment Transmission

In addition to interactiveness, investor sentient transmission also has the attribute of being multi-scale. Since sentiment transmission is, in nature, a kind of information transfer which is observed and studied at multiple temporal scales [35], it is undoubtedly significant to explore investor sentiment transmission at various scales. From an investor sentiment perspective, the reasons for the existence of multiple temporal scales can be explained by investor heterogeneity. Investor sentiment transmission exhibits heterogeneity based on heterogeneous investor belief theory. When the market implements short-sale constraints, optimistic emotions dominate the market and, thus, have a quick and wide diffusion which is observed at high frequencies (low scale or short term), while pessimistic investors do not participate in trading [36,37], which needs longer time to be diffused. Additionally, speculative investors and agents are motivated to own overpricing assets because they believe that they can resell them to optimistic players [38], which further accelerates the diffusion of optimistic emotions. This phenomenon leads to the asymmetric transmission that optimistic emotions are easily diffused in bullish markets, while pessimistic attitudes dominate the bearish markets.

To explore the multi-scale features of investor sentiment transmission, a wavelet method has been adopted. The wavelet method exhibits significant superiority in decomposing the time series into components at multiple frequencies [39]. For example, Chu et al. [40] use the wavelet method to obtain the causal relationship between stock return and investor sentiment. The results indicate a bi-directional non-linear causal relationship between return and sentiment after wavelet decomposition, while the original series manifest a uni-directional linear relationship. Dash and Maitra [41] adopt the wavelet method to obtain the decomposed series of sentiment and returns and compare the effect of sentiment on returns at multiple temporal scales. The study argues that the investment activities of short-term and long-term traders are both influenced by sentiment. Dash and Maitra [42] use the wavelet method to investigate the multi-scale sentiment correlation between the developed and emerging markets. They find that the sentiment of developed countries is always influential on that of emerging countries in the short and long run, while the sentiment of emerging countries affects that of developed countries only in the long run.

The above literature using wavelet and focusing on multiple temporal scales has indicated the implicit characteristics at different frequencies compared to the original series. As investor sentiment transmission also exhibits different features and patterns in the short and long term, capturing and discussing investor sentiment transmission at multi-scales can uncover the ‘hidden’ characteristics.

### 2.4. Research Gaps and Contributions

According to the review of related works, we find that current studies about investor sentiment transmission do not form a complete set of methods to estimate the investor sentiment transmission relationships. Our study fills the gap by applying a composite set of methods combining the wavelet, transfer entropy, and complex network analysis. These methods are selected and integrated based on the multi-scale, complex interactiveness, and directionality of investor sentiment transmission. This composite method can help to build a realistic transmission network of investor sentiment and reveal the local and network effect at multi-frequencies. In addition, this study is one of the few exploring the transmission patterns and characteristics of firm-specific investor sentiment. Current literature mainly underscores the relationships between investor sentiment and asset returns and ignores investor sentiment itself as a complex information interactive system. Our findings can provide strategic suggestions for heterogeneous investors and market regulators on paying attention to specific investor sentiment according to short-term or long-term objectives.

## 3. Data and Methodology

### 3.1. Data Description

We use daily data of the Baidu Search Index from 1 January 2015 to 31 December 2021 (2556 observations) to represent the investor sentiment of firms, with a focus on the 137 new energy vehicle (NEV)-related listed firms. The Baidu Search Index is the search volume of relevant keywords mentioning those 137 NEV-related listed firms, reflecting the investor sentiment on those firms. Baidu is the leading search engine used in China with a market share of over 85% at the end of 2021 (Source: https://gs.statcounter.com/search-engine-market-share/all/china (accessed on 12 August 2021)). Baidu Search Index is widely adopted as the proxy variable in research on investor sentiment in China [18,43].

### 3.2. Methodology

#### 3.2.1. The Overall Framework

Figure 1 shows the proposed framework for constructing the multi-scale ISTNs in this study. By analysing the networks, we can capture the transmission features of investor sentiment on different frequency scales among NEV-related firms. The time series of 137 firms’ Baidu Search Index serves as the proxy of investor sentiment. Using wavelet decomposition, we obtain two time series for each original series, namely time series of investor sentiment at high frequency and low frequency. Then, using the time series of investor sentiment at high frequency, we calculate the transfer entropy between any two firms which represents the transmission relationship of investor sentiment between these two firms. Further, we take those 137 firms as nodes and the estimated transmission relationships as edges between nodes to construct the ISTN at high-frequency. The same procedures are repeated for the investor sentiment at low frequency to obtain the ISTN at low frequency. At last, we conduct network analysis using network statistics and network motifs to investigate the transmission patterns of investor sentiment at high and low frequencies. The above steps are summarized as follows:Step 1: Multi-scale decomposition using the wavelet method;Step 2: Sentiment transmission based on transfer entropy;Step 3: Network construction of ISTNs on the high-frequency scale and low-frequency scale;Step 4: Network analysis using network statistics and motifs.

#### 3.2.2. Multi-Scale Decomposition

The wavelet method is a widely adopted and powerful theory for the multiple time scale analysis of the time series with non-linearity, proposed by Morlet in the 1980s. The wavelet method can effectively capture the local characteristics of a time series in both time and frequency dimensions, which has been applied in various areas, such as financial time series analysis.

In practical applications, the Continuous Wavelet Transform (CWT) and Discrete Wavelet Transform (DWT) are two typically used wavelet forms. DWT is the most applied to the multi-scale analysis of the financial time series, while the Maximal Overlap Discrete Wavelet Transformation (MODWT) is deemed an advanced version of DWT. For MODWT, a time series signal S(t) can be decomposed into a series of sub-sequences through two filters, namely the wavelet filter $h_{l}$ and the scaling filter $g_{l}$(l=0,⋯,L−1). $h_{i,l}$ and $g_{i,l}$ represent wavelet filter and scaling filter at the ith level. Then, the MODWT wavelet filter ${\tilde{h}}_{i,l}$ and the scaling filter ${\tilde{g}}_{i,l}$ at the *i*th level can be defined as
(1)$$\left\{\begin{array}{cc} {\tilde{h}}_{i,l} & =h_{i,l}/2^{i/2},\\ {\tilde{g}}_{i,l} & =g_{i,l}/2^{i/2}, \end{array}\right.$$
where
(2)$$\sum {\tilde{h}}_{i,l}^{2}=\sum {\tilde{g}}_{i,l}^{2}=\frac{1}{2^{i}}.$$

The wavelet and scaling coefficients, ${\tilde{W}}_{i}$ and ${\tilde{V}}_{i}$, can be defined as
(3)$$\left\{\begin{array}{cc} {\tilde{W}}_{i,t} & =\frac{1}{2^{i}/2}\sum_{l=0}^{L-1}{\tilde{h}}_{i,l}S\left(t-1\right),\\ {\tilde{V}}_{i,t} & =\frac{1}{2^{i}/2}\sum_{l=0}^{L-1}{\tilde{g}}_{i,l}S\left(t-1\right), \end{array}\right.$$
where t=1,⋯,N−1 and N is the length of S(t). ${\tilde{W}}_{i,t}$ and ${\tilde{V}}_{i,t}$ can also be written in a matrix form:(4)$$\left\{\begin{array}{cc} {\tilde{W}}_{i} & ={\tilde{w}}_{i}S,\\ {\tilde{V}}_{i} & ={\tilde{v}}_{i}S. \end{array}\right.$$

Finally, the original signal *S* can be decomposed by MODWT as:(5)$$S=\sum_{i=1}^{l}{\tilde{w}}_{i}^{T}{\tilde{W}}_{i}+{\tilde{v}}_{I}^{T}{\tilde{V}}_{I}=\sum_{i=1}^{I}{\tilde{D}}_{i}+{\tilde{A}}_{I},$$
where ${\tilde{D}}_{i}$ is the details of the original signal S(t) at scale *i* and ${\tilde{A}}_{I}$ is the smooth part of S(t) at scale *I*.

In our study, after MODWT, the original time series signal S(t) is decomposed into 5 sub-sequences at different scales as shown in Figure 2. D1 is at the highest frequency (1 day) and A4 is at the lowest frequency (more than 8 days). To explicitly present the differences between high-scale and low-scale investor sentiment transmission, we only choose D1 as the high-frequency series and A4 as the low-frequency series.

#### 3.2.3. Investor Sentiment Transmission

Transfer Entropy (TE) is an important measure of causality for stationary time series based on the information theory [32]. Inspired by the notion of Mutual Information (MI), TE between time series $X_{i},Y_{i},i=1,\cdots ,T$ is defined in the form of conditional MI following the research of Ma [44], as follows:(6)$$TE=\sum p\left(Y_{i+1},Y^{i},X_{i}\right)log\frac{p(Y_{i+1}|Y^{i},X_{i})}{p(Y_{i+1},Y^{i})},$$
where $Y^{i}=\left(Y_{1},\cdots ,Y_{i}\right)$ and *T* is the length of time series.

The above expression can be written as conditional MI, as follows:(7)$$TE=I(Y_{i+1};X_{i}|Y^{i}).$$

However, it has been widely considered that estimating MI is notoriously difficult. To find a supplementary measurement of MI, Copula Entropy (CE) is a recently introduced theory on the measurement of statistical independence [45]. It has been examined that CE is equivalent to MI. Specifically, MI can be represented by negative CE as below:(8)$$I\left(X\right)=-H_{c}\left(X\right),$$
where I(·) is MI and $H_{c}\left(\cdot \right)$ is CE.

Moreover, CE is an ideal measure of statistical independence for its properties, such as multivariate, symmetric, non-negative, invariant to monotonic transformation, equivalent to correlation coefficient in Gaussian cases. Therefore, we apply CE to calculate TE. The application of CE to calculate TE has been discussed by previous literature [44,46] and has been adopted by research, such as [47]. Here, we connect TE and CE through the expressions below. Specifically, starting from the TE definition, the TE equation can be transformed and divided into four CE parts as below:(9)$$\begin{array}{cc} TE & =\sum p\left(Y_{i+1},Y^{i},X_{i}\right)log\frac{p(Y_{i+1}|Y^{i},X_{i})}{p(Y_{i+1},Y^{i})}\\ \; & =\sum p\left(Y_{i+1},Y^{i},X_{i}\right)log\frac{p\left(Y_{i+1},Y^{i},X_{i}\right)p\left(Y^{i}\right)}{p\left(Y_{i+1},Y^{i}\right)p\left(Y^{i},X_{i}\right)}\\ \; & =I\left(Y_{i+1};Y^{i};X_{i}\right)-I\left(Y_{i+1};Y^{i}\right)-I\left(Y^{i};X_{i}\right)\\ \; & =-H_{c}\left(Y_{i+1};Y^{i};X_{i}\right)+H_{c}\left(Y_{i+1};Y^{i}\right)+H_{c}\left(Y^{i};X_{i}\right)\\ \; & =-H_{c}\left(Y_{i+1},Y^{i},X_{i}\right)+H_{c}\left(Y_{i+1},Y^{i}\right)+H_{c}\left(Y^{i},X_{i}\right)-H_{c}\left(Y^{i}\right), \end{array}$$
where $H_{c}\left(Y^{i}\right)=0$ if $Y^{i}=Y_{i}$.

With the above representation, we can calculate TE via only CEs, and CEs can be estimated based on the following steps. According to the nonparametric method for estimating CE (MI) proposed by Ma and Sun [45], CE is then estimated as follows.

Let X be random variables ($X={\left\{x_{1},\cdots ,x_{N}\right\}}^{T}$) with marginal distributions u and copula density c(u). CE of X is defined as
(10)$$H_{c}\left(X\right)=-\int_{u}c(u)logc(u)du.$$

To estimate c(u), we apply the following calculation of Empirical Copula Density (ECD). If given data samples $\left\{x_{1},\cdots ,x_{T}\right\}$ i.i.d. generated from random variables $X={\left\{x_{1},\cdots ,x_{N}\right\}}^{T}$, ECD can be calculated as follows:(11)$$F_{i}\left(x_{i}\right)=\frac{1}{T}\sum_{t=1}^{T}\chi (x_{t}^{i}\leq x_{i}),$$
where i=1,⋯,N and χ is the indicator function. Let $u=[F_{1},\cdots ,F_{N}]$, and then $\left\{u_{1},\cdots ,u_{T}\right\}$ is a sample as data from ECD c(u). Once c(u) is estimated, we can obtain CE using Equation (Equation 10). Then TE can be estimated as follows:(12)$$TE=-H_{c}\left(Y_{i+1},Y^{i},X_{i}\right)+H_{c}\left(Y_{i+1},Y^{i}\right)+H_{c}\left(Y^{i},X_{i}\right)-H_{c}\left(Y^{i}\right),$$
where $H_{c}\left(Y^{i}\right)=0$ if $Y^{i}=Y_{i}$.

Lastly, having obtained the values of TE between any two nodes, we conduct the significance test of TE at 95% level, which serves as the filtering process of investor sentiment transmission relationships. Insignificant TEs are removed, which indicates that there are no significant investor sentiment transmission relationships. The significant TEs are then used in the next step, that is, mapping the network. Here, in order to assess the statistical significance of TE estimates, we rely on a Markov block bootstrap as proposed by Dimpfl and Peter [48] and the corresponding tool is RTransferEntropy, an R package for transfer entropy. The Markov block bootstrap generates the simulated distribution of TE under the null hypothesis that information transfer equals zero, which remains the dependencies within each single time series but eliminates the dependencies between time series. Then, TE is estimated based on the simulated time series. In our paper, we did the bootstrap 1000 times to obtain the simulated distribution so that the significance test can be conducted. Every TE estimate, which represents the investor sentiment transmission relationship between two firms, is tested following the above procedures.

#### 3.2.4. Network Construction

The above methods have estimated and filtered the investor sentiment transmission relationships using transfer entropy. Here, we conduct Step 3, i.e., mapping the investor sentiment transmission networks (ISTNs). An ISTN contains two elements: firms as nodes and investor sentiment transmission relationships as edges, expressed as follows:(13)ISTN=(N,E),
where *N* is the set of nodes and *E* is the set of edges.

In our study, the nodes $N=\{v_{i};i=1,\cdots ,n\}$ are a set of firms where *n* is the total number of firms. The edges $E=\{e_{ij};i,j=1,\cdots ,n\}$ are the investor sentiment transmission relationships between firm *i* and *j* estimated by TE in Step 2. The matrix of the investor sentiment transmission relationships between firms can be expressed as follows:(14)$$E=\left[\begin{array}{ccc} e_{11} & \cdots & e_{1n}\\ \vdots & \ddots & \vdots \\ e_{n1} & \cdots & e_{nn} \end{array}\right],\left(i,j=1,\cdots ,n\right)$$

#### 3.2.5. Network Analysis

The structure of the ISTNs defines the function, i.e., transmission patterns. We can obtain the characteristics of transmission using several indicators. Our study uses the sum of edge weights, density, average weighted clustering coefficient, core number, betweenness, closeness, node degree, and node strength. The calculations and metrics are presented below in Table 1.

In addition, we conduct the motif analysis to explore the detailed transmission patterns of investor sentiment at high and low frequencies. Network motifs, as the basic components of complex networks, are a set of specific patterns of local interconnections with potential functional properties [49]. Those potential functions reflect specific forms of interactions among nodes, which, in the context of sentiment transmission, reflect the sentiment transmission activities among firms. To evaluate the main transmission patterns of investor sentiment, we calculate the probability that network motif $ms_{u}$ exists in network ISTN=(N,E) as below:(15)$$p\left(ms_{u}\right)=\frac{\sum_{i,j,k\in N}I(i,j,k,ms_{u})}{\sum_{i,j,k\in N}\sum_{ms_{u}\in S}I(i,j,k,ms_{u})},$$
where we incorporate 13 sets of motifs in S={S1,S2,⋯,S13} and $ms_{u}\in S$. $I(I,j,k,ms_{u})$ is the characteristic function. When the three nodes *i*, *j*, and *k* are connected in the pattern of $ms_{u}$, $I(I,j,k,ms_{u})=1$. Otherwise, $I(I,j,k,ms_{u})=0$.

Next, in order to see whether the main transmission patterns at high frequency are also the main patterns at low frequency, we estimate a sophisticated metric, that is, the conditional probability that one type of motif $ms_{u}$ exists in the low-frequency ISTN given the presence of the other motif $ms_{v}$ among the same group of nodes *i*, *j*, and *k* in the high-frequency ISTN. The specific definition is organized as below:(16)$$p\left(ms_{u}|ms_{v}\right)=\frac{\sum_{i,j,k\in N}I^{Hi}\left(i,j,k,ms_{v}\right)I^{Lo}\left(i,j,k,ms_{u}\right)}{\sum_{i,j,k\in N}\sum_{ms_{u}\in S}I^{Hi}\left(i,j,k,ms_{v}\right)I^{Lo}\left(i,j,k,ms_{u}\right)},$$
where $ms_{u},ms_{v}\in S,u,v=\left\{1,2,\cdots ,13\right\}$. $I^{Hi}\left(i,j,k,ms_{v}\right)$ is for the high-frequency ISTN (Hi means high), and $I^{Lo}\left(i,j,k,ms_{u}\right)$ is for the low-frequency ISTN (Lo means low).

## 4. Empirical Results and Discussion

Our study applied composite methods of wavelet, transfer entropy, and complex network to investigate the investor sentiment transmission at multiple temporal scales. According to the above steps of methodology and selections of network statistics, this paper obtains the daily data of the Baidu search index for 189 stocks comprising the new energy vehicle sector of China for 7 years from 2015 to 2021, and constructs a set of ISTNs. Each year, multi-scale extraction is carried out, respectively. We obtained the ISTNs at 5 different temporal scales according to the length of the time scale. In order to highlight the differences in time scales, this paper compares and shows the results on the short and long scales. The short scale represents a time scale of 1–2 days and the long scale represents a time scale of 16 days and longer. Our discussion is based on the ISTNs at high and low frequencies for each year from 2015 to 2021 and the corresponding network statistics.

### 4.1. Overall Network Structural Evolution

Figure 3 shows the overall network structural evolution from 2015 to 2021. We include four network statistics in the discussion, i.e., the sum of edge weights, density, average weighted clustering coefficient, and core number. Blue lines represent the structural evolution trend at high frequencies, while orange lines are at low frequencies.

Figure 3A shows the total edge weight of the ISTN at high and low frequencies. The total of edge weights represents the total amount of information transmission in the network, and the activity of transmission relationships or transmission behaviours in the ISTN. The blue line in the figure is higher than the yellow line as a whole, indicating that the investor sentiment transmission is more active at high frequencies.

Figure 3B shows the network density of the ISTN at high and low frequencies. The network density indicates the overall closeness of the nodes in the network measured by the fraction of existing edges to total potential edges. In ISTN, it indicates the overall closeness of the investor transmission relationships among firms. Generally, the overall closeness of investor transmission relationships under high frequency is higher. It further supports the results in Figure 3A that investor sentiment propagates more actively and frequently in the short term than in the long term.

Figure 3C shows the average weighted clustering coefficient of the ISTN at high and low frequencies. The average weighted clustering coefficient indicates the degree of interconnection between the neighbours of a node, which indicates the degree of closeness of local transmission relationships in the ISTN. First of all, it shows that the local transmission is stronger at high frequencies. Although we observe that the transmission of the entire network is stronger at high frequencies, it exhibits that the closeness of local transmission is larger than that of the entire network. In other words, short-term or irrational investor sentiment is easier to spread locally (within a small range).

Figure 3D shows the core number of the ISTNs at high and low frequencies. The core number represents the depth of network space or the order of the hierarchical structure (number of hierarchical levels) and the order or depth of the transmission relationships in the ISTN. As shown in the figure, the transmission path of investor sentiment at high frequency is deeper and farther, and the hierarchical structure of transmission is more obvious, which presents a stronger ‘pyramid tower’ structure. In other words, at a high frequency, the short-term or irrational investor sentiment of a firm at the top of the ‘pyramid tower’ needs to span longer hierarchical levels in order to finally affect the bottom of the tower. At low frequencies, there are fewer hierarchical levels so it is easier for the long-term rational emotions at the top of the tower to finally be transmitted to the bottom of the tower. In other words, in the short term, investor sentiment toward a firm will not spread to the whole system but diffuse around a relatively small range of ‘neighbours’, while, in the long term, it will have a systemic influence on the whole network.

Figure 3E shows the betweenness of the ISTNs at high and low frequencies. Betweenness measures the number of shortest pathways connecting any two nodes that pass through a certain node. In other words, betweenness is a way to detect the amount of influence a node has over the flow of information or investor sentiment in the network. When a node’s betweenness is high, it means that this node is an ‘intermediary’ in the ISTN. An ‘intermediary’ in the ISTN can serve as the bridge of information and investor sentiment, which can affect and even controls the information and investor sentiment from the sender nodes. As shown in the figure, the blue curve (high frequency) is above the yellow curve (low frequency) as a whole, which implies that the number of information intermediaries is more at high frequency than that at low frequency. Investor sentiment is distributed and transmitted through those intermediaries so that a firm’s investor sentiment can have higher possibilities to be transmitted to the whole network in the short term. This result reflects similar characteristics of investor sentiment transmission to the results of total edge weights that the transmission of investor sentiment is more active at high frequency.

Figure 3F shows the closeness of the ISTNs at high and low frequencies. Closeness is estimated by the reciprocal of the total length of the shortest paths between a certain node and all other nodes in the network. The larger the closeness of a node is, the closer it is to all other nodes and the more central it is in the network. This metric reflects the central position of a node in the whole network. As shown in the figure, the yellow curve (low frequency) is above the blue curve (high frequency) as a whole, which implies that at low frequency, the entities of ISTN are closer to each other on average. In other words, in the long term, generally, a firm’s investor sentiment can influence other firms’ investor sentiment through a relatively short path. This result is similar to that of the clustering coefficient that long-term sentiment is easier to spread systemically (within a wide range).

The above results have indicated the following findings. First, the transmission of investor sentiment is more active in the short term. Second, the investor sentiment toward a firm is prone to transmit locally in the short term and systemically in the long run. As demonstrated in the study of McClure et al. [50], passions or irrational emotions can dominate in short-run decision-making and rationality takes over in long-run decision-making. Yang and Gao [6] also indicate that irrationality has more significant influences on short-term decision-making. With the passage of time, more information related to the price and performance of firms is disclosed, and the influence of longer-term investor sentiment will become weaker than that of shorter-term investor sentiment. It can be inferred from our results that the transmission of irrational sentiment is more active in the short term, and becomes gradually weaker in the transmission process and, thus, is limited to a relatively local range. Thus, investors should pay more attention to the irrational sentiment if they are short-term arbitrageurs and more to the rational sentiment if they pursue long-term value investment [8].

### 4.2. Comparative Analysis of High-Frequency and Low-Frequency ISTNs

The above analysis has shown the network structural evolution at multi-scales (short term and long term) of investor sentiment transmission from 2015 to 2021 and has indicated the differences in sentiment transmission between the short and long term. To further explore the relations and differences between sentiment transmission in the short term and long term, we then have a comparative analysis from the level of nodes and edges, respectively.

From a node’s perspective, we can see whether the ‘celebrity’ of the sentiment transmission at high frequencies is also a ‘celebrity’ at low frequencies (shown in Figure 4). In ISTN, ‘celebrity’ refers to the node or firm with more transmission relationships [51]. Figure 4A identifies ‘celebrities’ by the order of centrality measured by degree among all nodes, and Figure 4B uses the weighted centrality measured by weighted degree. In Figure 4A, the *x*-axis is the rank of the node’s degree at high-frequency. The smaller the value of rank is, the larger the degree of the node is. The *y*-axis is the year from 2015 to 2021. The coloured area shows the rank of the node’s degree at low frequency. When a node with a high rank of degree at high-frequency is coloured in red, it means that this node also ranks high at low frequency. In Figure 4B, the rank is based on the node’s weighted degree (strength). When a node with a high rank of strength at a high frequency is coloured in red, it means that this node also ranks high at a low frequency. Figure 4A,B both reflect the comparison of the ‘celebrity effect’ between high and low frequencies, while Figure 4A focuses on the transmission range of a celebrity’s sentiment and Figure 4B focuses on the transmission strength of a celebrity’s sentiment. Although both reflect the characteristics of ‘celebrities’, the emphasis is different. The degree centrality reflects the number of transmission relationships or influence range of ‘celebrities’, while the weighted degree centrality reflects the intensity of transmission or influence strengths of ‘celebrities’. A ‘celebrity’ with a wide range of influence does not necessarily have a strong influence strength. For example, ‘celebrity *A*’ is known to many people, i.e., *A* has a wide range of influence, but people only have an ordinary or weak impression of *A*, that is, *A*’s influence strength is not high.

From Figure 4A,B, we can see that when a node is a ‘celebrity’ at high frequencies, it is also a ‘celebrity’ at low frequencies. However, the relationships reflected in Figure 4B are more explicit and strong than that in Figure 4A. In other words, nodes with greater transmission intensity at high frequencies also have greater influence strength at low frequencies. In the context of investor sentiment transmission, influence strength is expressed as the intensity or strength of the transmission relationship. The stronger the influence of firm *A* has, the stronger the transmission relationship of firm *A* is, which means the investor sentiment of firm *A* has a stronger impact on other firms.

Figure 5 discusses similar phenomena from an edge’s perspective. Figure 5A shows the ratio of overlapping investor sentiment transmission relationships at high and low frequencies, where the *x*-axis is the year from 2015 to 2021. The blue curve is the fitted trend when treating the network as a directed one while the orange curve is the fitted trend when treating the network as an undirected one, and the shaded areas under curves are confidence intervals at 95%. It can be observed that whether the direction of the transmission relationship is distinguished or not, the overlapping rate of the transmission relationships is about 46% to 54%, that is, about half of the transmission relationships exist at both high-frequency and low-frequency.

Figure 5B shows whether the strong transmission relationship at high frequency is also strong at low frequency. In Figure 5B, the rank is based on the edge’s strengths or edge weights. When an edge with a high rank of strength at a high frequency is coloured in red, it means that this edge also ranks high at a low frequency. In general, a strong transmission relationship at high frequency is also strong at low frequency. However, at high-frequency, the transmission relationships ranking in the last quarter show strong transmission strengths at low frequencies. It appears that some of the weak transmission relationships at high frequencies grow into strong relationships at low frequencies. Combining the results of Figure 5A that half of the transmission relationships exist both at low and high frequencies, we can infer that most of the transmissions have consistent strengths between the situations at high and low frequencies, which reflects the continuity of investor sentiment transmission. In addition, some of the transmissions have increasing strengths from short term to long term, which reflects the growth of investor sentiment transmission. It is plausible that the sentiments with growing strengths of transmission from short term to long term are those reflecting rational emotions and attitudes. While irrational emotions induced by rumours and scandals have strong transmissions among firms in the short term, rational emotions can make deviated sentiments return to normal levels. This process of rationality-reversion is implicit in the short term but is explicitly observed in the long term.

### 4.3. Analysis of ISTN with Motifs

Network motifs, in the context of sentiment transmission, reflect the sentiment transmission activities among firms. Thus, looking into the distribution of network motifs may provide a deep insight into the function of the entire network in terms of sentiment transmission. Figure 6 is the collection of network motifs.

Figure 7 shows the results of motif distribution at high and low frequencies. Figure 7A shows the distribution of each type of motif at high frequency. Wherein, the main motif is type S8. Considering the potential functions of motifs in the context of investor sentiment transmission, this paper believes that type S8 reflects a one-way (uni-directional) circulation of investor sentiment transmission. This motif shows that at high frequencies, investor sentiment transmission has strong causality and recursiveness.

Figure 7B shows the distribution of various motifs at low frequencies. Among them, the main motifs are type S5, S6, and S8. S5 and S6 reflect a multi-directional transmission structure driven by a core node. In S5, the driving node acts as the source of the sentiment transmission. These motifs show that at low frequencies, investor sentiment transmission is more flexible than that at high frequencies and exhibits some reciprocity which is driven by the transmission source. When sentiment transmission of the network is driven or initiated by several source firms, these source firms should be paid particular attention to. Thus, in the long run, we need to detect and monitor those driving firms of sentiment transmission. Their sentiments will have a larger influence on the entire network compared to other firms.

Figure 8 shows how the motifs in the short term change to another type in the long term. First, we observed that the evolution maintains high stability and consistency in different years (between 2015 and 2021), and forms several main evolution patterns from high to low frequencies. Taking 2015 as an example, it is mainly reflected in the evolution of the following motif pairs: (S7,S2), (S9,S5), (S12,S5), and (S13,S12). The former is the motif manifested at high frequencies, and the latter is at low frequencies. The pair reflects the evolution of motifs from short term to long term. These four motif pairs show the following two patterns (we do not discuss (S13,S12) since it only exhibits a mere reduction in random transmission with little information displayed).

(1) (S7,S2): From the outdegree structure at high frequency to the in-degree structure at low frequency. In the context of investor sentiment transmission, this evolution means that in the short term, the node that is easy to transmit its sentiment outward gradually becomes susceptible to other nodes in the long term. From another aspect, firms that are prone to transmit their irrational emotions to the outside in the short term will become more susceptible to the rational emotions of other firms in the long term. It is plausible that in the short term, irrational emotions toward a firm spread easily to other firms, but with the passage of time, affected firms will feed back rational emotions to the original source of irrational emotions through stable business performance and disclosure of favourable information.

(2) (S9,S5) and (S12,S5): From the cyclic structure at high frequency to the core node driving structure at low frequency. This evolution means that in the short term, the transmissions are diverse and active. With the enlarging of the time scale, some active transmission relationships gradually become silent and eventually form some dominant nodes. These dominant nodes play a driving role in the sentiment transmissions, that is, the emotions of other nodes need to pass through this driving node to further affect another node. In the short term, irrational investor sentiment tends to be disorderly transmitted and diffused. With the extension of time, the market no longer pays attention to and recognizes some irrational investor sentiment transmission relationships, but forms attention to some driving firms, which makes rational investor sentiment be transmitted orderly in the long run.

## 5. Conclusions and Perspectives

The aim of our study is to investigate the multi-scale transmission of investor sentiment based on complex network analysis. As investor sentiment transmission is drastically influential toward financial markets, prior studies do not reach a consensus in terms of the diverse and complex transmission patterns of investor sentiment on multi-scales. In our study, based on the complex interactiveness and multiple temporal scales (referred to as term structure otherwise) of investor sentiment transmission, we proposed a composite set of methods to explore the transmission patterns and characteristics of investor sentiment among firms.

After taking the Baidu search index data as the proxy for the investor sentiment toward firms, we use the wavelet method to extract multi-scale transmission information and adopt transfer entropy to measure the directional and non-linear sentiment transmissions among firms and construct investor sentiment transmission networks at high and low frequencies for each year from 2015 to 2021. By conducting complex network analysis, we find that the transmission of investor sentiment presents as more active in the short term and takes place in a local range. As irrational emotions dominate in short-term decision-making and rationality takes over for long-term decision-making, we suggest that investors pay more attention to the irrational sentiment factors if trading as short-term arbitrageurs and focus on rational sentiment factors as long-term value investors. On the other hand, we find that the transmission of investor sentiment presents patterns of continuity and growth from short term to long term. It suggests that the sentiment of some firms will have a growing influence in the long term despite that it is less perceived in the short term. Additionally, it is plausible that the growing sentiment of those firms will be a determining factor for macro economies and long-term investors. In addition, we also find that from short term to long term, the transmission patterns of investor sentiment will have specific evolution. It suggests that a firm’s emotions spread easily to other firms in the short term, but on a longer scale, affected firms will then have feedback on the previous source of emotions through stable performance and disclosure of favourable information. In addition, in the short term, investor sentiment is prone to be disorderly or diversely transmitted and diffused. While on a long scale, the market pays less attention to some irrational investor sentiment but more to the sentiment of some driving firms, which makes investor sentiment be transmitted orderly in the long term. It suggests that when investors pursue long-term benefits, they should detect those driving firms and have particular concentration on them. As for market regulators, those driving firms play an important role in the long-term stability of markets and economies. Regulators should monitor the investor sentiment toward those driving firms, which will provide a warning signal for systemic risk.

Overall, our study explores the multi-scale sentiment transmission patterns and characteristics and proposed a composite set of methods based on the complex interactiveness, causality of transmission and multiple temporal scales. While the transmission of investor sentiment presents discrepancies in different market norms and constraints, applying the composite method in the present study to different markets and countries for comparative analysis is the subject of ongoing research. In addition, since the time range used in our study is only 7 years, from 2015 to 2021, the findings are limited. First, the findings may not be effective outside this time range. Second, the fitted trends of the network statistics may be unsolid. Thus, it is also one of the future works to incorporate a longer time range to consolidate the findings.

## Figures and Tables

**Figure 1 entropy-24-01786-f001:**
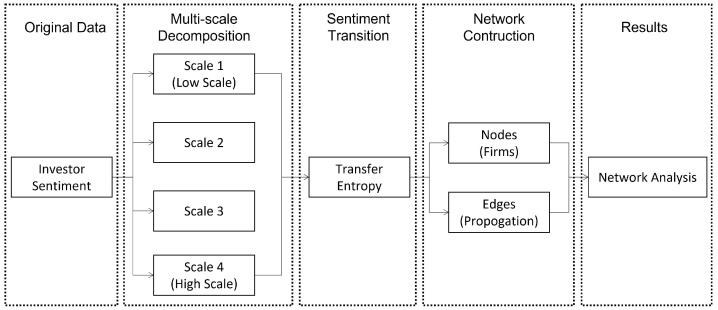
The framework of constructing multi-scale ISTNs.

**Figure 2 entropy-24-01786-f002:**
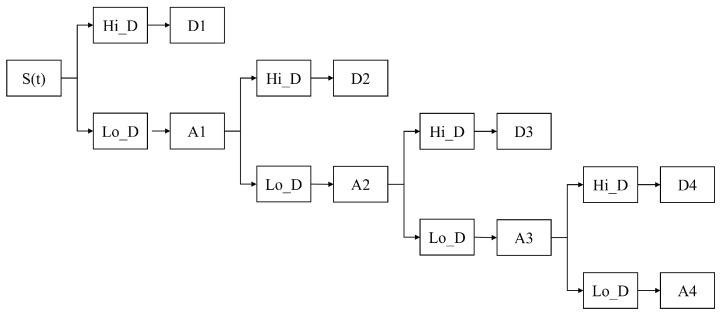
The process of decomposing the original signal S(t) at different scales.

**Figure 3 entropy-24-01786-f003:**
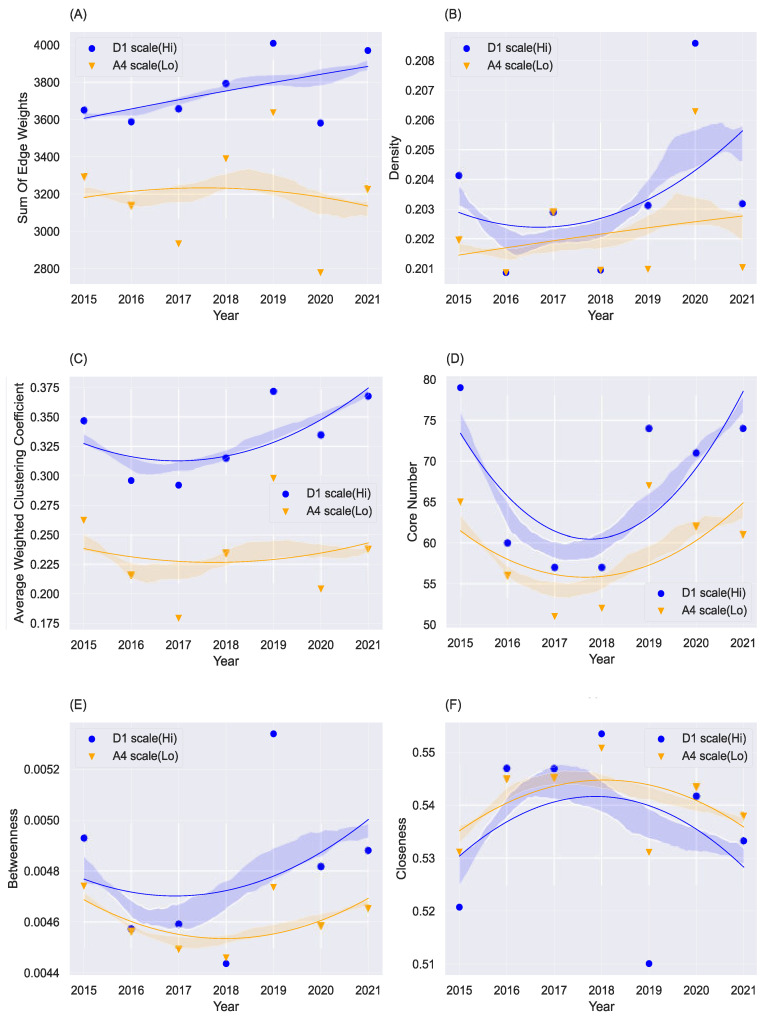
Overall network structure evolution at high and low frequencies from 2015 to 2021. The sum of edge weights, density, average weighted clustering coefficient, and core number on high- and low-frequency scales are compared in Subfigures (**A**–**F**) respectively. The *x*-axis is the year from 2015 to 2021. The points in subfigures are real values, and curves are fitted trends and the shaded areas under curves are confidence intervals at 95%.

**Figure 4 entropy-24-01786-f004:**
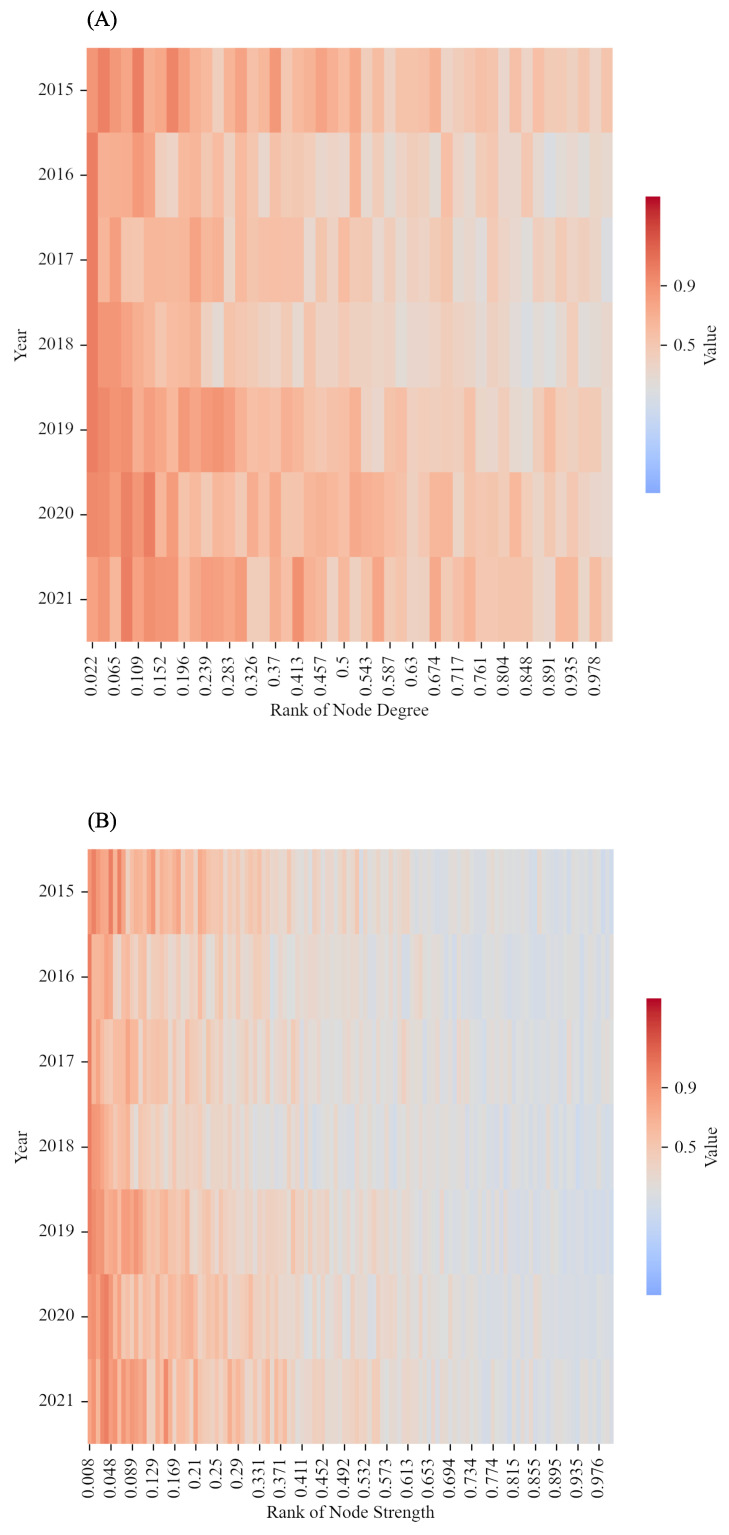
The comparison of ‘celebrity’ between high-frequency and low-frequency in each year from 2015 to 2021.

**Figure 5 entropy-24-01786-f005:**
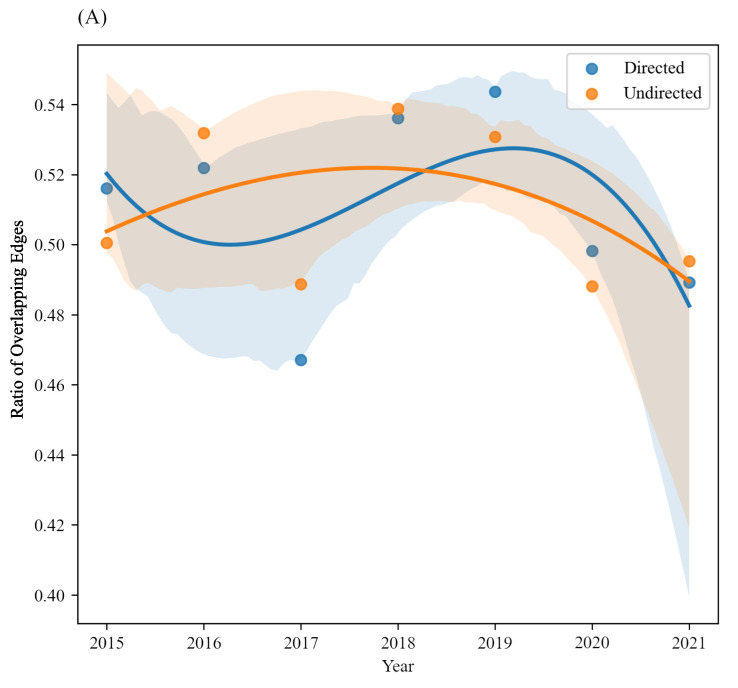
The comparison of ‘celebrity’ between high frequency and low frequency from 2015 to 2021 from the perspective of edge.

**Figure 6 entropy-24-01786-f006:**
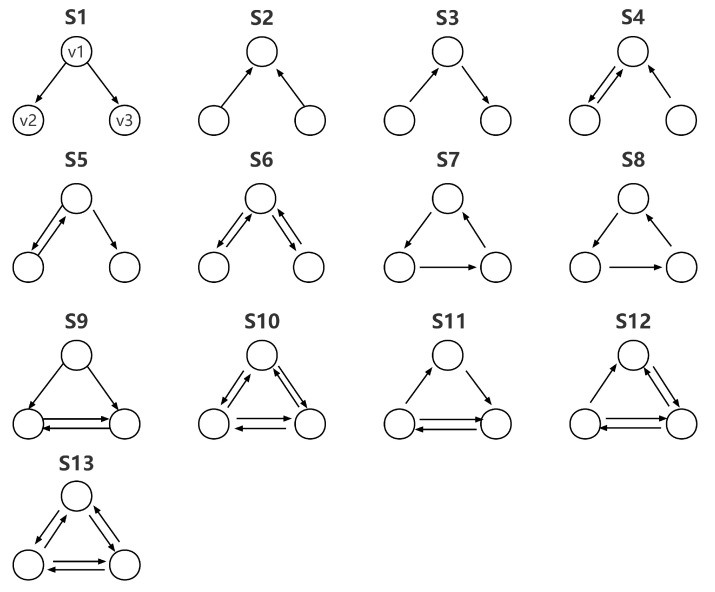
A collection of network motifs. Each motif consists of nodes v1, v2, and v3 and the interconnections between them. Since motifs have potential functional properties, when specific types of motifs take over the entire network, the function of the network will be influenced by the functions of these motifs.

**Figure 7 entropy-24-01786-f007:**
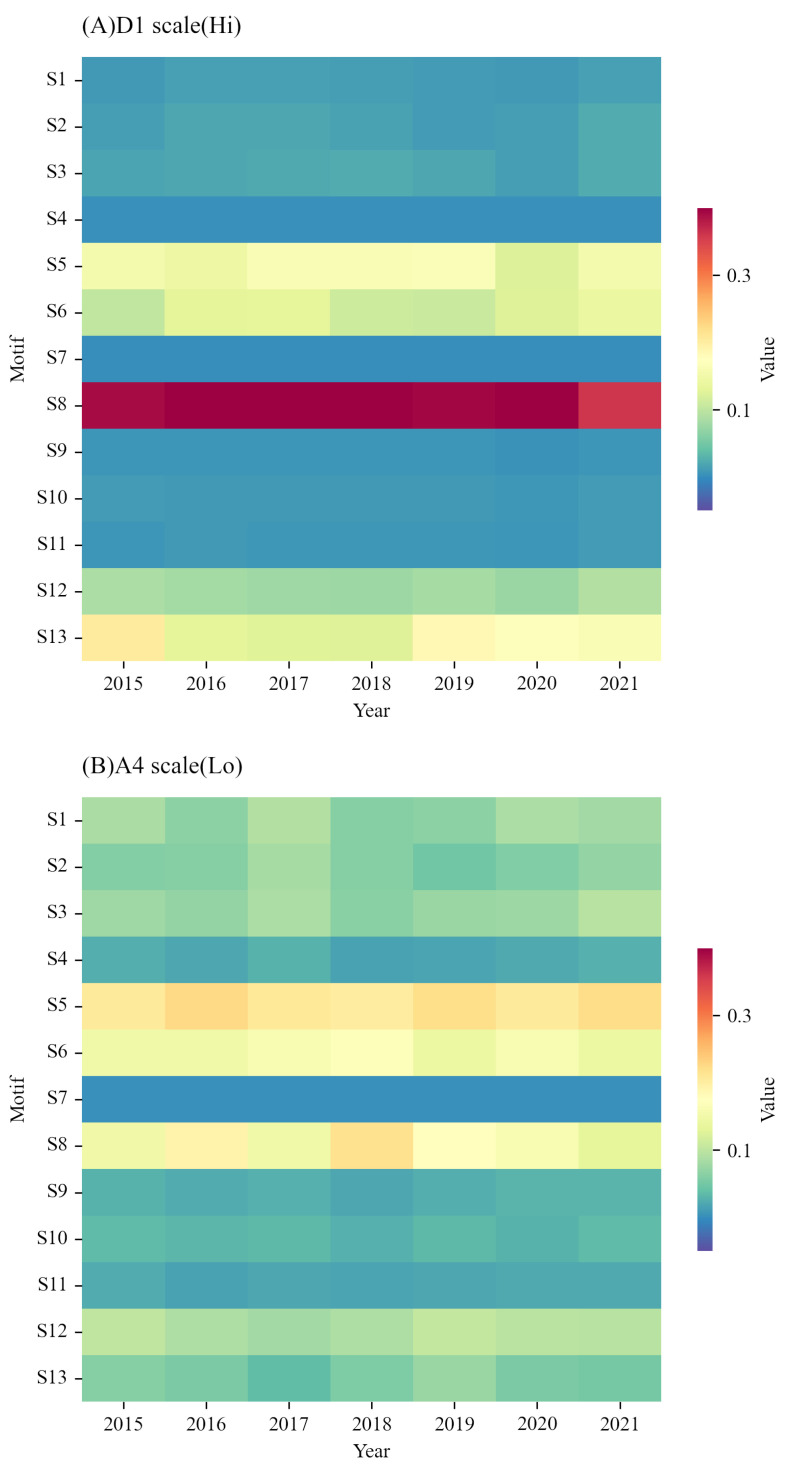
Distribution of network motifs at high and low frequencies. Subfigure (**A**) is the distribution of motifs at high frequency from 2015 to 2021 and Subfigure (**B**) is at low frequency. The red colour means that this type of motif appears a lot in the network. The *x*-axis is the year from 2015 to 2021 and the *y*-axis is the type of motif from S1 to S13. The definition of each type of motif is shown in Figure 6.

**Figure 8 entropy-24-01786-f008:**
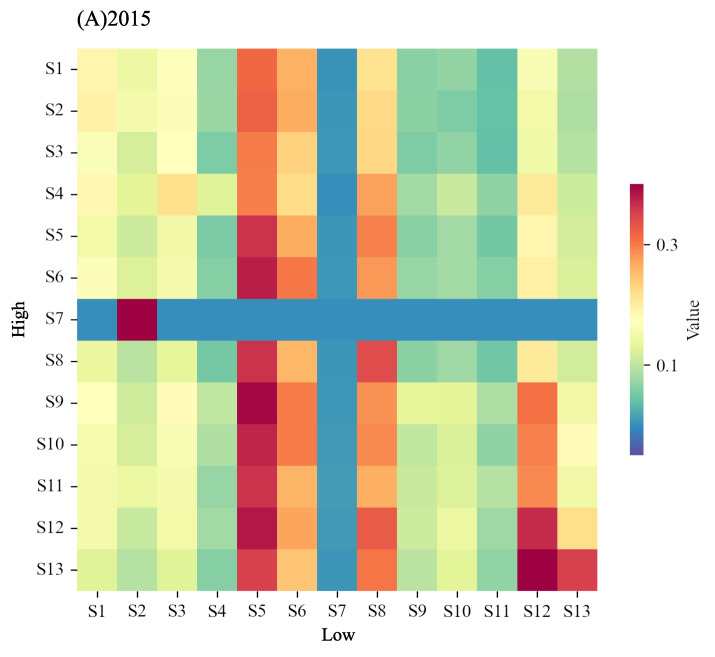
The evolution of motifs from high frequencies to low frequencies in 2015 and 2021. It reflects whether a specific type of motif at high frequencies changes to another type at low frequencies. The colour reflects the degree of such change. Subfigure (**A**) is the condition in 2015 and Subfigure (**B**) is in 2021.

**Table 1 entropy-24-01786-t001:** Network indicators used in our study.

Network Indicators	Descriptions
Sum of edge weights	It represents the total amount of information transmission in the network, and the activity of transmission relationships or transmission behaviours in the ISTN.
Density	It represents the overall closeness of the investor transmission relationships among firms.
Average weighted clustering coefficient	It represents the degree of interconnection between the neighbours of a node, which indicates the degree of closeness of local transmission relationships in the ISTN.
Core number	represents the depth of network space or the order of the hierarchical structure (number of hierarchical levels) and the order or depth of the transmission relationships in the ISTN.
Betweenness	measures the number of shortest pathways connecting any two nodes that pass through a certain node, which identifies the ‘intermediaries’ in the ISTN.
Closeness	is estimated by the reciprocal of the total length of the shortest paths between a certain node and all other nodes in the network, reflecting the central position of a node in the whole network.
Node degree	It represents the number of transmission relationships a node owns.
Node strength	It represents the total strength of transmission relationships a node owns.

## Data Availability

The data presented in this study are available on request from the corresponding author. The data are not publicly available due to privacy.

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
