# Peer review of "Multi-Scale Characteristics of Investor Sentiment Transmission Based on Wavelet, Transfer Entropy and Network Analysis"

_entropy, 2022, doi:10.3390/e24121786_

Round 1

Reviewer 1 Report

The paper is interesting to read and the results are quite nice. I have just several points that might/should be improved:

Major points:

I have not seen the application of CE to calculate TE, is there any relevant literature? If not, authors should do some sanity checks (e.g., compare this method with other estimators on particular examples) or use other estimators, e.g., Leonenko's estimator for transfer entropy.

Network measures: I would think about considering some additional measures, such as closeness and betweenness centrality, and some community detection algorithms and present the resulting network community structure. One can then also compare this structure to either heuristic classification (e.g., business sectors/subsectors) or other standard community detection methods.

Minor points:

Why there is a zero in the middle of the matrix in Eq. (12)?

Wrong reference (Fig. ??) - line 339

Figures - figures too small

Citations - check the right format

Reviewer 2 Report

the review report is attached 

Round 2

Reviewer 1 Report

My comments have been incorporated; the manuscript is now suitable for publication. 

Author Response

Thanks for all your kind reminders and suggestions.

Reviewer 2 Report

i would to thank the author for the extensive work provided after my revision.

I have only one minor comment:

1) the transfer entropy was firstly introduced by Thomas Schreiber and his paper on physical review letter should be at least mentioned. I have no issue with the author of the paper "Estimating transfer entropy via copula entropy" which in addition is not yet published. Please update the list of references.

Author Response

Thanks for all your kind reminders and suggestions. We cited Thomas Schreiber's paper in the newly revised manuscript. The track of changes in round 1 is coloured in BLUE and round 2 in RED. One change search for “HAN Round1” to see the round 1 changes in the PDF file and "HAN Round2" to see the round 2 changes.